# Language Models are Realistic Tabular Data Generators

**Vadim Borisov**[1,*] **Kathrin Seßler**[2,*] **Tobias Leemann**[1], **Martin Pawelczyk**[1], **Gjergji Kasneci**[1]
[1]University of Tübingen, Tübingen, Germany
[2]Technical University of Munich, Munich, Germany

## Abstract

Tabular data is among the oldest and most ubiquitous forms of data. However, the generation of synthetic samples with the original data's characteristics remains a significant challenge for tabular data. While many generative models from the computer vision domain, such as variational autoencoders or generative adversarial networks, have been adapted for tabular data generation, less research has been directed towards recent transformer-based large language models (LLMs), which are also generative in nature. To this end, we propose **GReaT** (**G**eneration of **Rea**listic **T**abular data), which exploits an auto-regressive generative LLM to sample synthetic and yet highly realistic tabular data. Furthermore, GReaT can model tabular data distributions by conditioning on any subset of features; the remaining features are sampled without additional overhead. We demonstrate the effectiveness of the proposed approach in a series of experiments that quantify the validity and quality of the produced data samples from multiple angles. We find that GReaT maintains state-of-the-art performance across numerous real-world and synthetic data sets with heterogeneous feature types coming in various sizes.

## 1 Introduction

Tabular data is one of the most common forms of data in machine learning (ML) – over 65% of data sets in the Google Dataset Search platform contain tabular files in either CSV or XLS formats (Benjelloun et al., 2020). However, due to the expensive nature of data collection processes, tabular data sets (i) are often class imbalanced, (i.e., tabular data sets tend to have long-tailed label distributions (Cao et al., 2019)), (ii) contain critical person-related information and cannot be shared due to privacy protection or socio-ethical principles (Gascón et al., 2017), and (iii) often come with impurity issues such as noisy or missing values which impede the application of modern ML algorithms (Lin & Tsai, 2020). Synthetically generated data has the potential to alleviate these three important issues. Therefore, the generation of realistic artificial tabular data has received considerable attention in recent years (Choi et al., 2017; Park et al., 2018; Xu et al., 2019; Borisov et al., 2021).

Apart from real-world impurity issues, there also exist various technical problems that make the generation of synthetic data difficult. Typically, tabular data contains various feature types, such as *categorical features* (e.g., `name`, `countryOfOrigin`, `jobTitle`) and *numerical features* (e.g., `age`, `income`).The categorical variables (i.e., words or clauses) may frequently contain most of the information. For example, the highly used Adult Income data set consists of seven numerical and eight categorical variables (Dua & Graff, 2017). This heterogeneity of feature types and values leads to three core challenges in tabular data preprocessing and modeling:

**Extensive and lossy preprocessing.** For most of the existing tabular data generation methods, extensive data preprocessing of tabular data is required, which usually includes the following steps: (i) categorical data encoding into numbers, (ii) data scaling or normalization, (iii) replacing missing values, and (iv) removing outliers and smoothing. These data transformation steps may result in the loss of important information or the introduction of artifacts that are not present in the original data. As an example, the categorical encoding into numeric values may introduce an artificial ordering into

---

[*]Equal contribution
Corresponding authors: kathrin.sessler@tum.de, vadim.borisov@uni-tuebingen.de

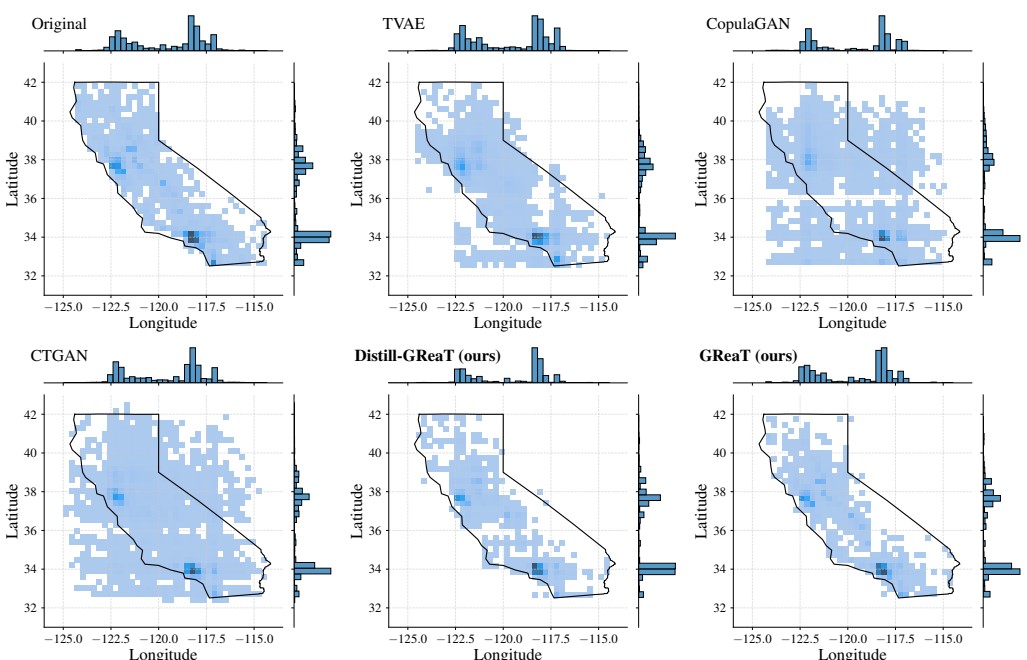

Figure 1: A comparison of the original and generated samples for the California Housing data set (Pace & Barry, 1997), which contains characteristic information about different properties in California, USA. We show joint histogram plots of the highly interconnected variables `Latitude` and `Longitude`. The black outline indicates the true boundary of the state of California.

previously unordered values (Borisov et al., 2021). Therefore, the problem of *lossy preprocessing* may strongly influence the quality of generated data as a result.

**Context knowledge for coherent semantics.** Almost all common synthetic data generation methods transform tabular data into a fully numerical representation. However, tabular data sets frequently consist of variables that are contextually interconnected. In the Adult Income data set (Dua & Graff, 2017), the features `age`, `marital-status`, and `education` have a clear coherence relationship: There is a certain minimal legal age for marriage, and it is challenging to get a Ph.D. at a young age. Such context knowledge should ideally be considered when generating *realistic* synthetic data samples. We refer to this common issue as the *contextual knowledge problem*.

**Arbitrary conditioning.** A versatile model that can generate data for a large variety of applications should be able to synthesize data conditioned on an arbitrary set of variables. This allows imputation of any missingness pattern in the data or oversampling of arbitrary subsets. Currently, the majority of the methods do not provide extensions to arbitrary conditioning and require the generative model to be re-trained according to each specific set of features to be conditioned on (Mirza & Osindero, 2014). We refer to generators that allow for conditional generation with any specified feature combination as supporting *arbitrary conditioning*.

Most modern deep-learning approaches for tabular data generation build on generative models transferred from the computer vision domain (Borisov et al., 2021), such as Variational Autoencoders (VAEs, Kingma & Welling, 2013) or Generative Adversarial Networks (GANs, Goodfellow et al., 2014). However, deep learning models have equally revolutionized the field of natural language processing (NLP, Radford et al., 2019; Brown et al., 2020; Raffel et al., 2020). Modern large language models (LLMs) are often constructed in the form of auto-regressive density models over sequences of words (Radford et al., 2019; Bengio et al., 2000). This begs the question to which extent successful architectures for NLP are apt to the tabular data generation task.

Carrying this thought further, we present a novel method for probabilistic data generation that covers the outlined core challenges and results in state-of-the-art performance (see Fig. 1 for a qualitative example). We argue that pretrained self-attention-based LLMs (Vaswani et al., 2017) are suitable

for the probabilistic modeling of heterogeneous tabular data sets after these data sets have been appropriately transformed into a *textual representation*. We do so by constructing syntactically correct sentences based on feature names and row values without losing information or introducing artificial orderings and thus mitigate the issue of *lossy preprocessing*. This step, to which we refer as *textual encoding*, maintains substantially more information than usual transformations. Since we include the variable names in the encoding, a model trained on this data can directly access contextual information. To be able to make sense of this information, we suggest using established and pretrained language models to perform the generation task. This could be a possible path towards tackling the *contextualization problem*. Finally, we introduce a feature order permutation step to shuffle the feature order in the textual encodings. Training a model on such data will result in a versatile generator that supports *arbitrary conditioning*. Specifically, our work offers the following contributions relative to the existing literature on the generation of synthetic tabular data:

- **Novel paradigm.** We propose the first approach for realistic heterogeneous tabular data modeling and generation utilizing a transformer-decoder network architecture. Thereby, *we connect tabular and textual data modalities via a textual encoding scheme*.

- **Arbitrary conditioning.** When trained on textual encodings with random feature order permutations, our model inherits its arbitrary conditioning power from the LLM, which can model the data distribution conditioned on any given subset of features and sample the remaining features.

- **Extensive experimental results.** We show that our **G**eneration of **Rea**listic **T**abular Data (GReaT) obtains state-of-the-art generative performance on a variety of data sets across several measures. We have open-sourced our experimental results, making them available as strong benchmarks for the benefit of the community.

- **Python package.** Finally, we provide an easy-to-use Python implementation of the GReaT model, where it takes only three lines of code to generate new synthetic samples. Access to the package is provided via `pip install be-great`.[1]

## 2 RELATED WORK

While the generation of images and text is extensively explored (Karras et al., 2020; Subramanian et al., 2017), the generation of synthetic tabular is less commonly considered in the recent machine learning literature. Classical methods for tabular data modeling include Bayesian networks, in particular those based on the Chow-Liu approximation (Chow & Liu, 1968) or statistical tools such as copulas (Kamthe et al., 2021). More recent methods for probabilistic tabular data modeling utilize generative adversarial networks (Choi et al., 2017; Park et al., 2018; Mottini et al., 2018; Xu et al., 2019; Koivu et al., 2020) or variational autoencoders (Xu et al., 2019; Ma et al., 2020; Vardhan & Kok, 2020; Darabi & Elor, 2021).

The mixed structure of discrete and continuous features along with their different value distributions still poses a significant challenge. CTGAN, the current state-of-the-art approach by Xu et al. (2019), places special focus on the conditional distributions between the features to generate semantically meaningful data. For non-Gaussian feature distributions, the authors propose a *mode-specific normalization* technique. However, the one-hot-encoding scheme used to encode the modes and the categorical features in this work can aggravate the "curse of dimensionality" problem (Bellman, 1966) in the presence of high-cardinality variables. Furthermore, it does not profit from contextual information. We thoroughly compare our method to their approach in the experimental evaluation section.

In a parallel development, the area of natural language processing was dominated by recurrent neural networks (RNNs) before self-attention-based neural networks (Vaswani et al., 2017) revolutionized this field. Based on their technique, various auto-encoding models (Devlin et al., 2018; Sanh et al., 2019; Lan et al., 2019) for tasks like sentence classification, sequence-to-sequence models (Raffel et al., 2020; Lewis et al., 2020) for translation or summarizing, and auto-regressive models (Radford et al., 2019; Brown et al., 2020) for natural language generation tasks showed the strength of the self-attention mechanism.

---

[1] https://github.com/kathrinse/be_great

In the light of these successes, transformer-based models have also been devised for tabular data classification (Arik & Pfister, 2019; Somepalli et al., 2021; Kossen et al., 2021) and learning joint representations of tabular and textual data (Yin et al., 2020). Padhi et al. (2021) investigated the generation of multi-variate time series data, using a transformer architecture based on BERT (Devlin et al., 2018). However, there are no prior works on generating highly realistic non-sequential tabular data with the help of attention-based LLMs. Our work is the first to rigorously explore this path, which leads to state-of-the-art performance.

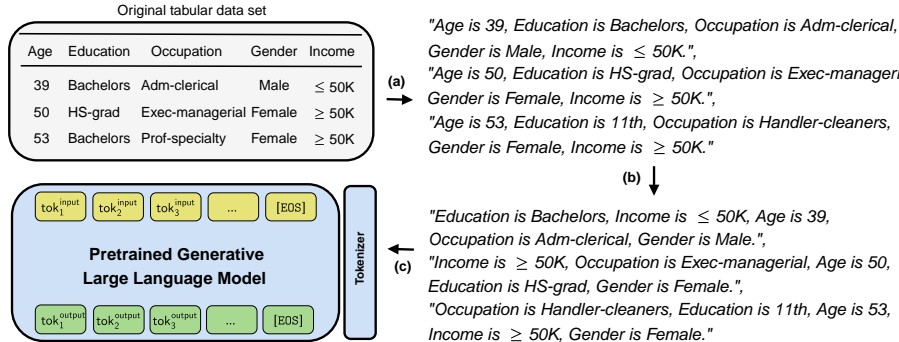

Figure 2: The GReaT data pipeline for the fine-tuning step. First, a textual encoding step transforms tabular data into meaningful text (a). Subsequently, a feature order permutation step is applied (b), before the obtained sentences can be used for the fine-tuning of a large language model LLM (c). The toy tabular data set inspired by the Adult Income data set (Dua & Graff, 2017).

## 3    GReaT: Generation of Realistic Tabular Data

This section presents the GReaT approach for fully-conditional tabular data generation using transformer-based neural networks. GReaT consists of two major stages: (1) the fine-tuning of a pretrained large language model (LLM) on a textually encoded tabular data set as shown in Fig. 2 and (2) sampling from the fine-tuned LLM to generate synthetic tabular data. We illustrate the sampling procedure in Fig. 3. In the following, we describe each component of the fine-tuning and sampling steps in detail. We conclude this section with a brief summary of our approach.

### 3.1    GReaT fine-tuning

**Textual encoding.** Standard pretrained generative LLMs expect sequences of words as inputs. Hence, to apply an LLM on tabular data, we have to convert each row of our data set into a textual representation. To this end, we propose the textual encoder.

**Definition 1 (Textual encoder)** *Given a tabular data set of $m$ columns with feature names $f_1, f_2, \ldots, f_m$ and $n$ rows of samples $\mathbf{s}_1, \ldots, \mathbf{s}_n$, we let the entry $v_{i,j}, i \in \{1, ..., n\}, j \in \{1, ..., m\}$ represent the value of the $j$-th feature of the $i$-th data point. Taking the feature name and value into account, each sample $\mathbf{s}_i$ of the table is transformed into a textual representation $\mathbf{t}_i$ using the following subject-predicate-object transformation:*

$$t_{i,j} = [f_j, \text{“is”}, v_{i,j}, \text{“,”}] \qquad \forall i \in \{1, ..., n\}, j \in \{1, ..., m\}, \qquad (1)$$
$$\mathbf{t}_i = [t_{i,1}, t_{i,2}, ..., t_{i,m}] \qquad \forall i \in \{1, ..., n\}, \qquad (2)$$

*where $t_{i,j}$, the textually encoded feature, is a clause with information about a single value and its corresponding feature name, and $[\,\cdot\,]$ denotes the concatenation operator.*

*Remark 1.* For the scope of this work, we treat the target variable as a regular feature in our formulations.

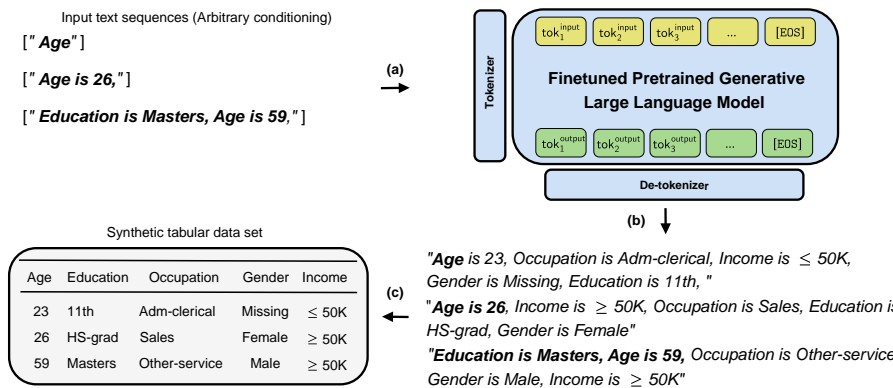

Figure 3: The sampling procedure of the proposed method for synthetic data generation. In order to generate new data points using a pretrained LLM, it is necessary to transform either a single feature name or an arbitrary combination of feature-value pairs into text (a). Subsequently, the input is completed by the fine-tuned LLM (b) and can be transformed back into a tabular format (c). In comparison to other state-of-the-art approaches, GReaT allows arbitrary conditioning on feature subsets without model retraining, i.e., the sampling can be performed by conditioning on any feature name or combination of feature names and values.

Fig. 2 (upper panels) shows an illustrative example of the proposed encoding technique. This result could be a sequence like *"Occupation is doctor, Gender is female, Age is 34,"*, which requires minimal preprocessing and does not suffer from information loss. While in standard natural language sequences the word order is crucial, in our case there is no preferred order between the individual features.

**Random feature order permutation.** By transforming a tabular feature vector into a sequence using the textual subject-predicate-object encoding scheme, pseudo-positional information is artificially introduced into the transformed tabular data sample. However, naturally there is no spacial ordering relationship between features in tabular data sets (Borisov et al., 2021). To reconstruct the feature order independence, we randomly permute the encoded short sentences $t_{i,j}$ of the full textual representation $\boldsymbol{t}_i$ by permutations $\boldsymbol{k}$.

**Definition 2 (Feature order permutation function)** *Formally, the result of applying a random feature order permutation $\boldsymbol{k}$ to $\mathbf{t}_i$, where each $k_j \in \{1, ..., m\}$ is arbitrary and $k_j \neq k_{j'}$ for $j \neq j'$, is defined as $\mathbf{t}_i(\boldsymbol{k}) = [t_{i,k_1}, t_{i,k_2}, ..., t_{i,k_m}] \quad \forall i \in \{1, ..., n\}$.*

When using shuffled orders of the textually encoded features, we fine-tune our generative language model on samples *without order dependencies*. Moreover, using permutations of the above form is highly beneficial as they allow for *arbitrary conditioning* in tabular data generation, which is further discussed in the following Subsection 3.2.

**Fine-tuning a pretrained auto-regressive language model.** Finally, we describe the fine-tuning procedure of a pretrained LLM to the encoded tabular data for generation tasks. We suppose a textually encoded tabular data set $\mathcal{T} = \{\mathbf{t}_i(\boldsymbol{k}_i)\}_{i=1,...,n}$ that was transformed into text by the proposed encoding scheme. Let $\boldsymbol{k}_i$ be randomly drawn permutations and $n$ denote the number of rows.

To be processed with an LLM, the input sentences $\mathbf{t} \in \mathcal{T}$ need to be encoded into a sequence of tokens from a discrete and finite vocabulary $\mathcal{W}$. These tokens can be character, word or subword encodings such as the Byte-Pair-Encodings (Sennrich et al., 2015). Thus, $\boldsymbol{t} \in \mathcal{T}$ is represented by a sequence of tokens $(w_1, \ldots, w_j) = \text{TOKENIZE}(\mathbf{t})$ with tokens $w_1, \ldots, w_j \in \mathcal{W}$, where $j$ denotes the number of tokens required to describe the character sequence $\mathbf{t}$. Commonly, the probability of natural-language sequences is factorized in an auto-regressive manner in LLMs (Jelinek, 1980; Bengio et al., 2000). It is represented as a product of output probabilities conditioned on previously observed tokens,

$$p(\boldsymbol{t}) = p(w_1, \ldots, w_j) = \prod_{k=1}^{j} p(w_k | w_1, ..., w_{k-1}). \tag{3}$$

In this formulation, it becomes evident that LLMs formally need to be highly capable predictors for follow-up tokens given an arbitrary-length sequence of preceding tokens. Indeed, the model is trained to output a probability distribution over possible next tokens $w_k$ from an input sequence $w_1, ..., w_{k-1}$ of arbitrary length. The entire model is usually fitted by optimizing the parameters to maximize the probability $\prod_{t \in \mathcal{T}} p(t)$ of the entire training data set.

As a result, an end-user can choose any existing *generative language* model for tabular data modeling and exploit the vast amount of contextual knowledge present in these models (Roberts et al., 2020). For instance, in generative transformer-decoder LLM architectures (e.g., the GPT models (Radford et al., 2018; 2019; Brown et al., 2020)) word-embeddings are obtained from large corpus of text (e.g., GPT3 model trained on 45TB of textual data (Brown et al., 2020)). By learning from such large collection of data, Transformers are able to build robust contextualized representations of language (Liu et al., 2021). Fine-tuning enables the model to leverage this contextual information in combination with the feature and category names to boost the models capabilities in a manner that is similar to transfer learning by learning bidirectional representations (Raffel et al., 2020).

## 3.2 GReaT SAMPLING OF SYNTHETIC DATA

Having obtained a fine-tuned auto-regressive model $q$ of the textual training data set that returns a categorical output distribution $z = q(w_1, \ldots, w_{k-1})$ over possible follow up tokens for an input sequence $w_1, \ldots, w_{k-1}$, we can apply several sampling strategies. Usually, the next token $\omega$ is sampled by weighted choice sampling with a temperature parameter $T > 0$ from the output $z$ of the LLM,

$$p(\omega | w_1, \ldots, w_{k-1}) = \frac{e^{(z_\omega/T)}}{\sum_{\omega' \in \mathcal{W}} e^{(z_{\omega'}/T)}}. \quad (4)$$

We note that the auto-regressive paradigm offers the possibility of sampling from token distributions $p(w_{k+1:j}|w_{1:k})$ with arbitrary preconditioning $w_{1:k}$. When we use random feature order permutations at train time, we can also start the textual sequence with any possible combination of features and values at inference time. As a result, the GReaT method is particularly flexible and could possibly be used in a variety of real-world problems such as missing value imputation (Kachuee et al., 2020) or generation of realistic counterfactual explanations (Pawelczyk et al., 2020). Moreover, the sampling of the conditional distribution comes at no additional cost.

**Sampling and extraction of synthetic tabular data.** Therefore, the setup provides several ways to sample new tabular data points using the GReaT method. We initialize the model with certain conditions and let the LLM sample the remaining tokens to complete the feature vector (in its textual representation). In total, we propose three options of preconditioning:

- **Feature Name Preconditioning.** Only a feature name, but no value is provided as an initial condition. This type of conditioning is able to generate samples from the entire joint data distribution $p(V_1, \ldots, V_n)$, where $V$ denotes the random variables representing the $n$ features in the data set.

- **Name-Value Pair Preconditioning.** In this case, a single feature name and a value are provided. Starting from this input, GReaT will complete the sample. This approach will generate samples from the distribution $p(V_{\setminus\{i\}}|V_i = v_i)$. Because modeling a single feature distribution is usually tractable (by frequentist estimation for categorical features or by fitting a parametric density estimator for continuous features), we can first sample a value for $V_i$ and then apply Name-Value Pair Preconditioning to sample the remaining features. Thereby, we can also sample the entire data distribution.

- **Multiple Name-Value Pair Preconditioning.** We can also provide multiple Name-Value pairs $V_{i_1} = v_{i_1}, \ldots, V_{i_k} = v_{i_k}$ as precondition to the LLM to realize arbitrary conditioning. By providing the textual encoding of this condition, we are able to sample from the distribution of the remaining features effectively $p(V_{\setminus\{i_1, \ldots, i_k\}}|V_{i_1} = v_{i_1}, \ldots, V_{i_k} = v_{i_k})$.

We illustrate the GReaT sampling possibilities in Fig. 3 and Fig. 8 that underline the high flexibility of the proposed approach for synthetic data generation. We apply commonly accepted pattern-matching algorithms using regular expressions to convert the generated textual feature representations back to a tabular format (Aho, 1991). We dismiss the respective samples in the rare case where

Figure 4: Distance to closest record (DCR) distributions for the California Housing data set with respect to the original train set. "Original Test Data Set" shows the DCR between the original test set and the original train set. This experiment shows that the proposed method does not "copy" samples from the training set but rather generates new synthetic samples close to the original samples.

the required format is violated. Generation rates of invalid samples were monitored and found to be consistently below 1 %. The main cause of this rare event lies in the generation procedure of the pre-trained LLM, which chooses the next token based on a probability distribution. For example, the `Occupation` variable in the Adult dataset has a specific support, e.g., {"Adm-clerical", "Exec-managerial", "Prof-specialty", ... }. However, in rare cases, the pre-trained LLM might leave this support and output, for example, 'Adm clerical' (without dash), or mix up these categories and return 'Adm-managerial'. To lower the chances of this unwanted effect, we suggest reducing the sampling temperature $T$ (Eq. 4), which sharpens the probability distribution. In practice, a simple validation function can quickly reject those infrequently invalid samples.

**A brief summary of the strengths of the GReaT method.** GReaT comes with several significant advantages over related approaches: It (i) allows the end-user to have full probabilistic control over the sampling procedure by its arbitrary conditioning power; it (ii) utilizes the knowledge from large text data bases to obtain a better representation that includes context knowledge; (iii) the proposed approach is easy to use, since it does not require a preprocessing of the data values. There is no need to specify discrete or numerical variables beforehand and the information loss due to preprocessing is therefore kept at its bare minimum.

## 4 EXPERIMENTAL EVALUATION

In this section, we empirically demonstrate the performance of the proposed GReaT approach using multiple qualitative and quantitative experiments. Lastly, for better reproducibility, we provide information on the packages and parameters for the selected LLMs.

**Data sets.** For the evaluation of the proposed algorithm, we utilize six real-world data sets that come from various domains. They also come in different sizes, reaching from less than 1,000 to over 100,000 samples. We also consider three synthetic data sets with varying numbers of features. Key characteristics of each data set are presented in Table 7. We split all data sets into 80% train and 20% test sets to avoid any data leakage. All models are trained or fine-tuned on the same training data samples. To demonstrate the power of our synthetic data generation framework to work out-of-the-box with real-world data, we apply zero data preprocessing, e.g., feature names and values are used as they are presented in original data sets. In contrast, other baselines require extensive data preprocessing.

**Baselines.** As baselines for our experiments, we use three deep learning-based methods. CT-GAN (Xu et al., 2019) is based on a generative adversarial network (GAN) (Goodfellow et al., 2014) for tabular data that allows to condition the generation process on only a single discrete feature. The same authors proposed TVAE (Xu et al., 2019), a variational autoencoder (VAE) for tabular data. The CopulaGAN model from the Synthetic Data Vault (SDV) framework (Patki et al., 2016) is applying the Gaussian copulas to simplify the underlying CTGAN. We analyze the main properties of the chosen baselines in Tab. 10.

We compare those baselines to the proposed method with two different pretrained transformer-decoder LLM models of various sizes. The smaller, distilled version of GPT-2 (Sanh et al., 2019) has 82 million learned parameters. We term the corresponding tabular data generator Distill-GReaT. An original version of GPT-2 by Radford et al. (2019) has over 355 million trainable parameters – we refer to this version simply as GReaT. A description of the architecture of the selected generative LLMs can be found in Table 8. We apply name-value pair preconditioning to start sampling.

For the evaluation of synthetic tabular data, we select four measures, all of which have been featured in multiple previous studies on synthetic tabular data generation (Borisov et al., 2021).

**Machine learning efficiency (MLE).** Since the generated data set should be able to replace the real data in a training process, this measure evaluates the performance of discriminative models trained on synthetic data sets. Therefore, the models are tested on real test data, and the scores are compared to the original performance when the models were trained on the original, real training data set. Table 1 and Table 4 present the results for the machine learning efficiency of proposed generative models compared to the baseline models. To be independent of the concrete discriminative model, we evaluated the synthetic data sets using a Linear/Logistic Regression (LR), a Decision Tree (DT) and a Random Forest (RF) (Ho, 1995). We observe that either GReaT and Distill-GReaT outperform all competitors and entail considerable performance improvements.

| Dataset | | Original | TVAE | CopulaGAN | CTGAN | Distill-GReaT | GReaT |
|---|---|---|---|---|---|---|---|
| | LR | 82.72±0.00 | 79.58±0.00 | 73.30±0.00 | 73.30±0.00 | 78.53±0.00 | **80.10±0.00** |
| Travel (↑) | DT | 89.01±0.00 | 81.68±1.28 | 73.61±0.26 | 73.30±0.00 | 77.38±0.51 | **83.56±0.42** |
| | RF | 85.03±0.53 | 81.68±1.19 | 73.30±0.00 | 71.41±0.53 | 79.90±0.53 | **84.30±0.33** |
| | LR | 96.69±0.00 | 94.70±0.00 | 94.57±0.00 | 94.44±0.00 | 96.56±0.00 | **97.72±0.23** |
| Sick (↑) | DT | 98.94±0.29 | 95.39±0.18 | 93.77±0.01 | 92.05±0.41 | 95.39±0.44 | **97.72±0.23** |
| | RF | 99.28±0.06 | 94.91±0.06 | 94.57±0.01 | 94.57±0.00 | 97.72±0.10 | **98.30±0.13** |
| | LR | 71.80±0.00 | 71.04±0.00 | 42.03±0.00 | 57.72±0.00 | 70.58±0.00 | **71.90±0.00** |
| HELOC (↑) | DT | 81.90±0.02 | 76.39±0.10 | 42.36±0.10 | 61.34±0.09 | **81.40±0.15** | 79.10±0.07 |
| | RF | 83.19±0.21 | 77.24±0.25 | 42.35±0.34 | 62.35±0.35 | **82.14±0.13** | 80.93±0.28 |
| | LR | 85.00±0.00 | 80.53±0.00 | 80.61±0.00 | 83.20±0.00 | 84.65±0.00 | **84.77±0.00** |
| Adult Income (↑) | DT | 85.27±0.01 | 82.80±0.08 | 76.29±0.06 | 81.32±0.02 | 84.49±0.04 | **84.81±0.04** |
| | RF | 85.93±0.11 | 83.48±0.11 | 80.46±0.21 | 83.53±0.07 | 85.25±0.07 | **85.42±0.05** |
| | LR | 58.76±0.00 | 56.34±0.00 | 40.27±0.00 | 50.93±0.00 | 57.33±0.00 | **57.34±0.00** |
| Diabetes (↑) | DT | 57.29±0.03 | 53.30±0.09 | 38.50±0.02 | 49.73±0.02 | 54.10±0.04 | **55.23±0.04** |
| | RF | 59.00±0.08 | 55.17±0.10 | 37.59±0.31 | 52.23±0.17 | 58.03±0.16 | **58.34±0.09** |
| | LR | 0.40±0.00 | 0.65±0.00 | 0.98±0.00 | 0.61±0.00 | 0.57±0.00 | **0.34±0.00** |
| California Housing (↓) | DT | 0.32±0.01 | 0.45±0.01 | 1.19±0.01 | 0.82±0.01 | 0.43±0.01 | **0.39±0.01** |
| | RF | 0.21±0.01 | 0.35±0.01 | 0.99±0.01 | 0.62±0.01 | 0.32±0.01 | **0.28±0.01** |

Table 1: ML efficiency experiment. The best results are marked in **bold**, the second-best results are underlined. Six real-world data sets are used, Travel Customers, Sick, HELOC, Adult Income, Diabetes, and California Housing. Each data set is evaluated on three discriminative ML models, Linear/Logistic Regression (LR), Decision Tree (DT), and Random Forest (RF). For classification tasks the accuracy score is reported, in the case of regression the mean squared error is used. Results are averages over five trials with different random seeds (cf. Appendix B for additional measures).

| | CopulaGAN | TVAE | CTGAN | Distill-GReaT | GReaT |
|---|---|---|---|---|---|
| Travel | 78.19±0.33 | 72.80±0.50 | **71.26±0.40** | 85.84±0.43 | 78.68±0.26 |
| Sick | 92.32±0.16 | 99.76±0.03 | 99.99±0.03 | 73.00±0.24 | **63.77±0.31** |
| HELOC | 97.83±0.12 | 100.00±0.00 | 99.99±0.01 | **68.30±0.47** | 69.15±0.36 |
| Adult Income | 88.54±0.09 | 88.49±0.18 | 97.23±0.10 | 69.79±0.17 | **62.84±0.08** |
| Diabetes | 99.85±0.01 | 98.31±0.03 | 87.84±0.14 | 99.82±0.01 | **72.29±0.10** |
| California Housing | 85.48±0.16 | 85.04±0.27 | 82.98±0.27 | 76.18±0.15 | **70.68±0.30** |
| Average | 90.37±0.03 | 90.73±0.03 | 89.88±0.03 | 78.82±0.05 | **69.57±0.05** |

Table 2: Discriminator measure (accuracy in %). Lower accuracy values indicate that the discriminator cannot differentiate between fake records and real samples. The accuracy of a perfectly indistinguishable data set would be 50%. Best results are **bold**, second-best results are underlined. Results are averages over five trials with different random seeds.

**Distance to closest records (DCR) histogram.** To verify that the generated data is similar to original samples while not being exact copies, this measure computes the distance to the closest record in original training data set $\mathcal{T}_{train}$. For each synthetic record $s_{gen}$, it is given by $\mathrm{DCR}(s_{gen}) = \min\{\mathrm{Distance}(s_{gen}, s_i) | s_i \in \mathcal{T}_{train}\}$. As a distance measure, we use the $L_1$-norm of the differences. We set the difference to be 0 for equal for categorical features and to 1 otherwise. In the best case, all DCR scores are non-zero and their distribution is close to that of DCRs computed with points from the original test data set $\mathcal{T}_{test}$. Visualizations of the distribution of the minimal distances can be found in Fig. 4 and Appendix B. All utilized models generated unseen samples in close proximity to the originals, and no significant difference was observed.

| Distill-GReaT | Adult | | HELOC | | Calfornia | | Travel | |
|---|---|---|---|---|---|---|---|---|
| | discr. (↓) | MLE (↑) | discr.(↓) | MLE (↑) | discr. (↓) | MLE (↓) | discr.(↓) | MLE (↑) |
| w/o permutation | **61.18±0.16** | **85.71±0.07** | 79.18±0.23 | 81.97±0.55 | 76.47±0.26 | **0.26±0.01** | 61.62±0.67 | **83.77±0.33** |
| with permutation | 69.77±0.09 | 85.25±0.07 | **68.29±0.34** | **82.14±0.13** | **76.09±0.14** | 0.33±0.01 | 85.65±0.38 | 80.31±0.53 |
| w/o pretraining | 99.14±0.02 | 84.15±0.05 | 99.51±0.01 | 76.32±0.20 | 85.10±0.22 | 0.34±0.01 | **62.98±0.71** | 81.47±0.42 |
| with pretraining | **69.77±0.09** | **85.25±0.07** | **68.29±0.34** | **82.14±0.13** | **76.09±0.14** | **0.33±0.01** | 85.65±0.38 | 80.31±0.53 |

Table 3: Results of experiments with and without permutation, as well as with and without pretraining based on discrimination (discr.) and ML efficiency (MLE, measured by the accuracy of a Random Forest model) on four real-world data sets. In all experiments, we used Distill-GReaT with the same training and pretraining setup; the only difference is the input data and pretraining. Results are averages over five trials with different random seeds.

**Discriminator measure.** To check whether the generated data cannot be easily told apart from the original data, we train a Random Forest discriminator (with hyperparameter tuning) on a mix of the generated train set (with label 0) and the original train set (with label 1). We then report the test accuracy on a test data set, which contains equal shares of samples from the generated test set and the real test set. Scores are shown in Table 2 and demonstrate the superior performance of the GReaT approach, which on average (across all used data sets) decreases the discriminator performance by 16.2 % over other competitive baselines for tabular data generation.

**Bivariate joint distribution plots.** We qualitatively compare the generated feature distributions by the baselines and the GReaT approach to that of the original data. As an illustrative example, we present joint density plots for the `Longitude` and `Latitude` features in the California Housing data set in Fig. 1. While CTGAN and CopulaGAN fail to model the strong dependency between these variables (indicated by out-of-distribution samples having both high latitude and longitude or low values for both), Distill-GReaT and GReaT yield densities that align well with the ground truth boundaries. TVAE shows mediocre performance but still places density mass outside the bounds.

**Effects of pretraining and permutations.** Having obtained impressive results with the complete setup, we investigate the role of the individual components towards its success. We compare the full model (with permutation and pretraining step) to a model without permutation and a model without pretraining. Tab. 3 presents ML efficiency and discriminator results for the modified models. On all but the very small "Travel" data set, we observe pretraining to help boost the performance. This might be due to the context knowledge made available to GReaT through the extensive adaptation to large text corpora. The results regarding the feature order permutation step are mixed – the MLE performance decreases with permutations, but the results for the discriminator metric are inconclusive. With permutations, the learning problem is undoubtedly more demanding. It provides models with the ability to perform generation based on arbitrary conditioning. However, we conjecture that it might also increase performance in some cases because it does not introduce any possibly unnatural, fixed feature ordering into the tabular data.

## 5 CONCLUSION

In our recent work, we investigate how state-of-the-art generative language models can be leveraged to synthesize highly realistic tabular data samples. Instead of following the usual path of encoding heterogeneous tabular data in a numerical format, we devise a textual encoding strategy and represent it with sentences that capture each record's semantics. The resulting transformer-decoder network fine-tuned on this data exhibits unprecedented generative performance and outstanding flexibility at the same time. We term our method **G**eneration of **Rea**listic **T**abular data (GReaT). GReaT unites several remarkable characteristics that address key problems in tabular data modeling: First, minimal preprocessing is required, which is efficient and results in the least possible information loss, thereby tackling the issue of possibly *lossy preprocessing*. Second, leveraging random feature order permutations, we exploit the capability of *arbitrary conditioning* and are thus equipped with full control over the probabilistic sampling procedure for tabular data. Finally, through pretraining, we can incorporate *contextual and semantic knowledge* extracted from terabytes of textual data for a more authentic tabular data synthesis. In conclusion, we see our work as a door opener leading to yet-undiscovered possibilities in the domain of heterogeneous data generation.

ETHICS STATEMENT

Various critical applications rely on heterogeneous tabular data (e.g., healthcare, finance, and so forth) and would profit from free data sharing. However, this always requires ethics and privacy considerations. The present work proposes a novel synthetic tabular data generation approach based on pretrained large language models. Before sharing any data (including data generated by the proposed method), the authors thus strongly encourage possible owners of proprietary data sets to verify that reverse identification is impossible or is prevented by regulatory means. Other than that, we see no further ethical questions related to this work.

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

## A  DISCUSSION

**Processing of numerical values with LLMs in the GReaT approach.** Since we convert heterogeneous tabular data into text (Sec. 3), continuous and discrete numerical values are represented as character sequences. While this might seem counter-intuitive at first, multiple independent studies have shown that transformer-based models are capable of understanding and processing numerical data encoded in such a way (Wallace et al., 2019; Brown et al., 2020). This is even true in the one-shot setting for inference tasks (Dinh et al., 2022). The impressive performance of GReaT that encompasses the numerical features aligns with these previous observations. Having said that, smarter encodings for numerical values can also be considered a possible path to further improvement.

**Unification of multiple modalities through transformer models.** Along with the usual numerical values, tabular data frequently contains textual metadata, e.g., feature names, named categories ("male", "female"), and open text features (e.g., `remarks`). The recent evolution of transformer neural networks now permits to holistically unite this information, which was processed separately in the past, and learn context-specific, robust, and meaningful representations. In the case of tabular data, information comes from the textual and numerical data modalities. The GReaT approach and the results presented in this work provide an initial clue of the range of possibilities and opportunities that may lie in this line of research.

## B  ADDITIONAL EXPERIMENTAL RESULTS

### B.1  FURTHER MLE MEASURES

Next to the accuracy measures (Tab. 1) we also report the ROCAUC score and the F1 score for the ML efficiency experiment (Sec. 4).

| | | Original | | TVAE | | CopulaGAN | | CTGAN | | Distill-GReaT | | GReaT | |
|---|---|---|---|---|---|---|---|---|---|---|---|---|---|
| | | ROCAUC | F1 | ROCAUC | F1 | ROCAUC | F1 | ROCAUC | F1 | ROCAUC | F1 | ROCAUC | F1 |
| TR (↑) | LR | 81.90±0.00 | 81.84±0.00 | 80.14±0.00 | 78.73±0.00 | 60.78±0.00 | 62.00±0.00 | 77.70±0.00 | 62.00±0.00 | 79.61±0.00 | 77.44±0.00 | **81.47±0.00** | **79.53±0.00** |
| | DT | 95.74±0.00 | 88.34±0.00 | **83.32±1.93** | 81.78±1.20 | 50.59±0.48 | 62.73±0.60 | 56.89±0.00 | 62.00±0.00 | 71.50±0.62 | 77.01±0.45 | 80.35±0.13 | **83.63±0.38** |
| | RF | 94.15±0.62 | 84.68±0.49 | 87.25±0.45 | 80.45±1.18 | 51.56±0.31 | 62.00±0.00 | 66.64±1.12 | 65.37±1.35 | 87.90±0.32 | 79.79±0.48 | **89.90±0.30** | **84.33±0.37** |
| SI (↑) | LR | 95.03±0.00 | 96.49±0.00 | **94.94±0.00** | 93.65±0.00 | 77.76±0.00 | 91.93±0.00 | 75.95±0.00 | 91.86±0.00 | 94.34±0.00 | 96.29±0.00 | 93.47±0.00 | **96.74±0.00** |
| | DT | 96.68±1.30 | 98.96±0.28 | 81.43±0.85 | 95.06±0.18 | 66.93±0.00 | 91.53±0.00 | 61.26±0.22 | 90.83±0.22 | 91.12±0.94 | 97.63±0.18 | **92.09±1.09** | **97.79±0.19** |
| | RF | 99.78±0.04 | 99.32±0.08 | 94.25±0.44 | 93.72±0.15 | 72.52±2.07 | 91.93±0.00 | 71.25±4.13 | 91.93±0.00 | 96.71±0.23 | 97.54±0.20 | **97.37±0.23** | **98.32±0.14** |
| HE (↑) | LR | 79.43±0.00 | 71.79±0.00 | 77.05±0.00 | 71.01±0.00 | 43.04±0.00 | 41.90±0.00 | 62.73±0.00 | 57.63±0.00 | 77.44±0.00 | 70.59±0.00 | **78.88±0.00** | **71.47±0.00** |
| | DT | 89.52±0.04 | 81.81±0.03 | 82.56±0.18 | 76.39±0.10 | 35.98±0.20 | 42.12±0.11 | 62.18±0.12 | 61.24±0.09 | **89.10±0.11** | **81.40±0.15** | 88.80±0.13 | 78.68±0.07 |
| | RF | 90.52±0.13 | 83.15±0.20 | 85.29±0.20 | 77.20±0.25 | 38.60±0.35 | 42.27±0.33 | 65.34±0.10 | 62.29±0.35 | **89.81±0.10** | **82.12±0.13** | 89.07±0.09 | 80.71±0.29 |
| AD (↑) | LR | 90.48±0.00 | 84.55±0.00 | 87.15±0.00 | 81.38±0.00 | 81.92±0.00 | 79.80±0.00 | 87.86±0.00 | 83.19±0.00 | 89.52±0.00 | **84.60±0.00** | **90.25±0.00** | 84.52±0.00 |
| | DT | 89.60±0.05 | 84.31±0.01 | 84.68±0.14 | 82.36±0.07 | 73.41±0.06 | 76.22±0.04 | 84.47±0.06 | 80.76±0.02 | **88.20±0.12** | 83.57±0.05 | 88.07±0.12 | **84.29±0.04** |
| | RF | 91.45±0.04 | 85.21±0.10 | 88.73±0.07 | 83.44±0.09 | 77.53±0.19 | 79.11±0.17 | 88.47±0.06 | 83.00±0.08 | 90.54±0.03 | 84.57±0.10 | **90.77±0.06** | **84.85±0.04** |
| DI (↑) | LR | 65.62±0.00 | 54.61±0.00 | 60.35±0.00 | 52.93±0.00 | 50.12±0.00 | 36.70±0.00 | 57.24±0.00 | 48.43±0.00 | 63.39±0.00 | 54.18±0.00 | **64.19±0.00** | **54.42±0.00** |
| | DT | 62.91±0.02 | 53.35±0.01 | 56.98±0.04 | 51.82±0.03 | 50.88±0.02 | 31.17±0.03 | 55.52±0.04 | 47.49±0.03 | 59.66±0.05 | 51.36±0.03 | **61.27±0.03** | **53.16±0.02** |
| | RF | 65.02±0.08 | 53.87±0.10 | 60.07±0.08 | 52.11±0.07 | 49.03±0.27 | 30.41±0.45 | 57.09±0.14 | 49.20±0.17 | 62.72±0.12 | 52.80±0.17 | **63.75±0.12** | **54.20±0.15** |

Table 4: Additional results of the ML efficiency experiments measuring the ROCAUC score and the F1 score for the five real-world classification data sets, Travel Customers (TR), Sick (SI), HELOC (HE), Adult Income (AD) and Diabetes (DI). Each data set is evaluated on three discriminative ML models, Linear/Logistic Regression (LR), Decision Tree (DT), and Random Forest (RF). The best results are marked in **bold**, the second-best results are underlined. Results are averages over five trials with different random seeds.

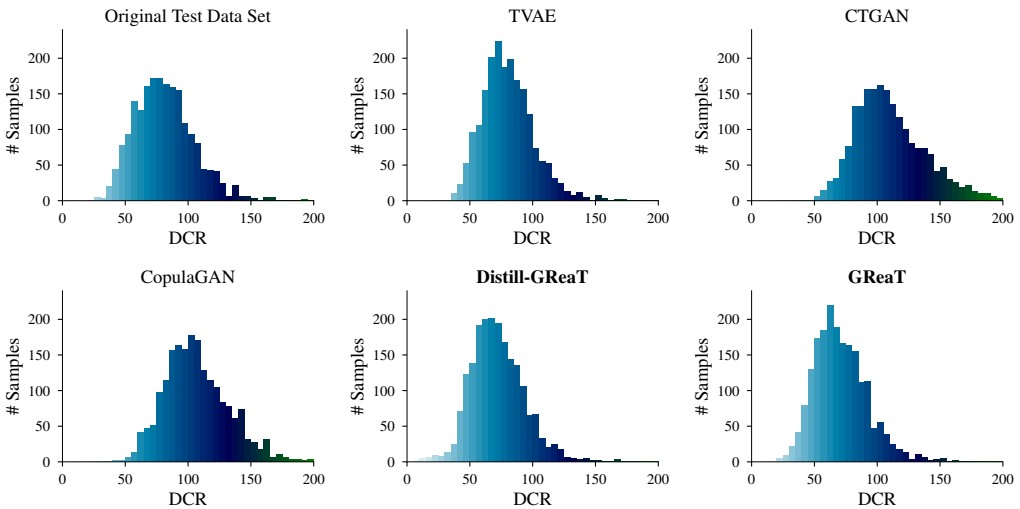

Figure 5: Distance to closest record (DCR) distributions for the HELOC Data set with respect to the original train set. "Original Test Data Set" shows the DCR between the original test set and the original train set. According to this experiment, the proposed method does not "copy" samples from the training set but rather generates new synthetic samples close to the original samples.

## B.2 AVERAGE NEGATIVE LOG-LIKELIHOOD METRIC FOR SYNTHETIC DATA

The generated data should be likely under the training data's distribution. Therefore, we generate three common synthetic data sets, and compute both the likelihood of the sampled data under the training data's density ($\mathcal{L}_{syn}$). This can however be prone to overfitting, which is why we additionally deploy the likelihood fitness metric proposed by Xu et al. (2019) ($\mathcal{L}_{test}$). To compute this metric, a parametric density model (BNs and GMMs respectively) is fitted to the *generated* data. Then the likelihood of the original test samples is computed. The results in Table 5 indicate that LLMs are comparable with state-of-the-art deep neural networks when modeling high-dimensional mixture distributions.

## B.3 DISTANCE TO CLOSEST RECORD RESULTS

We compare the distribution of the minimal distances of the generated samples to the training data set. Figure 4 shows the distribution for the California Housing data set and Figure 5 for the HELOC

| | GMM | | Asia (BN) | | Alarm (BN) | |
|---|---|---|---|---|---|---|
| Model | $\mathcal{L}_{syn}$ | $\mathcal{L}_{test}$ | $\mathcal{L}_{syn}$ | $\mathcal{L}_{test}$ | $\mathcal{L}_{syn}$ | $\mathcal{L}_{test}$ |
| Identity | -4.403 | -4.403 | -2.265 | -2.265 | -11.739 | -11.739 |
| CopulaGAN | -4.406 | **-4.676** | -6.321 | -3.129 | -22.438 | -16.828 |
| TVAE | **-4.205** | -4.817 | -2.418 | **-2.289** | -11.919 | **-12.074** |
| CTGAN | -4.427 | -4.723 | -3.999 | -2.528 | -20.804 | -15.248 |
| Distill-GReaT | -4.372 | -5.124 | **-1.925** | -2.355 | **-6.941** | -12.689 |

Table 5: Average Log-likelihood of synthetic data samples on a density model derived from the original data ($\mathcal{L}_{syn}$) and of the original test data on the model derived from the syntethic data ($\mathcal{L}_{test}$). The best results are marked in **bold**, the second-best results are underlined.

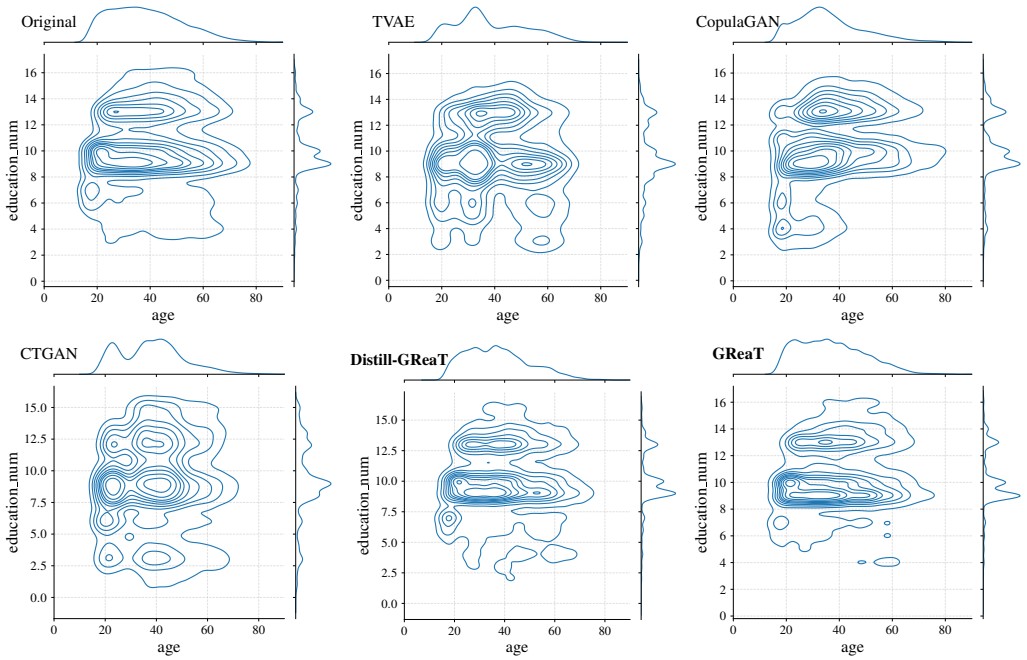

Figure 6: A comparison of the original and generated data sets for the Adult Income data set using joint plots for two related features – `Age` and `EducationNum`.

data set. The results indicate that the generated samples are close to the original ones without coping them exactly.

## B.4 ADDITIONAL QUALITATIVE ANALYSIS RESULTS

In Figure 6 we additionally compared the joint feature distribution of two exemplary features from the Adult Income data set, `Age` and `EducationNum`. The joint plots were computed using a kernel density estimator.

## B.5 RUNTIME COMPARISON

We also analyse the training/fine-tuning and sampling time between baseline models and two versions of the GReaT method using only two GPUs (Sec. C). Table 6 summarizes our time-benchmarking results. To make a fair comparison, we used the latest available versions of the corresponding implementations. Also, we utilize the same DL framework - PyTorch (Paszke et al., 2019), and the same number of epochs 200 as well as the sampling size - 1000, for each model.

While the proposed model requires significantly more time for the fine-tuning step compared to other baselines, Distill-GReaT and GReaT models demonstrate top performance for the vast majority of benchmarks across various datasets (Tab.1, Tab.2, Tab. 4). Also, there are critical applications such as healthcare (Hernandez et al., 2022) or finance (Assefa et al., 2020), where high quality synthetic tabular data is required, regardless of the possible computation cost, therefore in our experiments we introduced two data sets from medical machine learning field (Tab. 8). Lastly, with more computation resources, the proposed model's sampling and fine-tuning times can be drastically decreased.

|  | TVAE | CopulaGAN | CTGAN | Distill-GReaT | GReaT |
|---|---|---|---|---|---|
| training / fine-tuning time | 0:46 min | 2:30 min | 1:10 min | 1:35 h | 9:10 h |
| sampling time | 0.28 sec | 0.119 sec | 0.045 sec | 4 sec | 17 sec |

Table 6: A run time comparison of all generative models of our study. Selected models were trained/fine-tuned for 100 epochs and 1000 samples were generated.

|  | Dataset | Domain | #Samples | #Num | #Cat | Task | #Classes |
|---|---|---|---|---|---|---|---|
| (TR) | Travel Customers | Churn | 954 | 2 | 4 | Classification | 2 |
| (SI) | Sick | Medical | 3772 | 7 | 22 | Classification | 2 |
| (HE) | HELOC | Financial | 9,871 | 21 | 2 | Classification | 2 |
| (AD) | Adult Income | Social | 32,561 | 6 | 8 | Classification | 2 |
| (DI) | Diabetes | Medical | 101,766 | 8 | 39 | Multi-Class | 3 |
| (CH) | California Housing | Real Estate | 20,640 | 8 | 0 | Regression | - |
|  | Alarm | Synthetic | 20,000 | 0 | 37 | - | - |
|  | Asia | Synthetic | 20,000 | 0 | 8 | - | - |
|  | GMM | Synthetic | 6,000 | 2 | 0 | - | - |

Table 7: Details of the real-world and synthetic data sets used in the experimental evaluations. #Num and #Cat columns indicate numbers of numerical and categorical features in each data set.

|  | #Parameters | #Layers | #Heads | Embedding Size | Context Size |
|---|---|---|---|---|---|
| Distill GPT-2 | 82M | 6 | 12 | 768 | 1024 |
| GPT-2 | 355M | 24 | 16 | 1024 | 1024 |

Table 8: Structural details about the pretrained large language models used in our study.

## C  REPRODUCIBILITY DETAILS

**Reproducibility details.** We utilize pretrained generative language models from the established HuggingFace framework (Wolf et al., 2020). Its routines are also used for the fine-tuning and sampling steps. We open-sourced our implementation.

| | LR | DT | RF | |
|---|---|---|---|---|
| | max_iter | max_depth | max_depth | n_estimators |
| Travel Customers | 100 | 6 | 12 | 75 |
| Sick | 200 | 10 | 12 | 90 |
| HELOC | 500 | 6 | 12 | 78 |
| Adult Income | 1000 | 8 | 12 | 85 |
| Diabetes | 500 | 10 | 20 | 120 |
| California Housing | - | 10 | 12 | 85 |

Figure 7: The hyperparameter configuration of the evaluation models for the ML efficiency experiments.

We fine-tune the Distill-GReaT model for each data set for 200 epochs, except for the California housing and Diabetes (Strack et al., 2014) data sets, for them, we fine-tune them for 100 epochs. The GReaT baseline is fine-tuned for 110, 310, 400, 255, 150, 85, epochs for California Housing, Adult Income, Travel, Home Equity Line of Credit (HELOC), Sick (Dua & Graff, 2017), and Diabetes data sets, respectively. Depending on the GPU memory limitations, we vary the batch size from 8 to 124. For the sampling step, we set the temperature

| | |
|---|---|
| Travel Customers | `https://www.kaggle.com/datasets/tejashvi14/tour-travels-customer-churn-prediction` |
| Sick | `https://www.openml.org/search?type=data&sort=runs&id=38&status=active` |
| HELOC | `https://www.kaggle.com/datasets/averkiyoliabev/home-equity-line-of-creditheloc` |
| Adult Income | `https://archive.ics.uci.edu/ml/datasets/Adult/` |
| Diabetes | `https://www.kaggle.com/c/1056lab-diabetes-readmission-prediction` |
| California Housing | `https://www.kaggle.com/datasets/camnugent/california-housing-prices` |

Table 9: URLs for real-world data sets of the study.

parameter $T$ to 0.7 for all experiments and data sets. We sample new synthetic data using the name-value pair preconditioning (Sec. 3), starting with the target feature for all data sets (see an example in the supplementary materials).

We utilize the AdamW optimizer (Loshchilov & Hutter, 2017) for the proposed generative models, with the learning rate $5 \times 10^{-5}$. We plan to share trained weights for the GReaT models. The baseline models are trained for 200 epochs for each data set.

Our hardware setup consisted of two NVIDIA 2080RTX GPUs with 12 GB RAM each, 126 GB system RAM, and AMD Ryzen 3960X with 24 cores, we use the Ubuntu 20.04 operation system.

For the ML efficiency and discriminator experiments (Sec, 4) we additionally use linear/logistic regression, decision tree, and random forest models from the Scikit-Learn package (Buitinck et al., 2013), we report the exact hyperparameters for each model in Table 7. For the discriminator measure experiment (Table 2), we tune hyperparameters for each data set using the 5-fold cross-validation.

Besides providing important information for the experiment reproducibility, we also share the synthetic data sets generated by our GReaT method.

| | Single-Variable Conditional Sampling | Multi-Variable Conditional Sampling | Usage of the Context (Variable Names) | Transfer Learning | Data Prepossessing |
|---|---|---|---|---|---|
| CopulaGAN | ✓ | ✓ | ✗ | ✗ | Scaling, Encoding |
| TVAE | ✗ | ✗ | ✗ | ✗ | Scaling, Encoding |
| CTGAN | ✓ | ✗ | ✗ | ✗ | Scaling, Encoding |
| **GReaT** | ✓ | ✓ | ✓ | ✓ | Format conversion |

Table 10: An analysis of the main properties of the synthetic tabular data generation frameworks
.

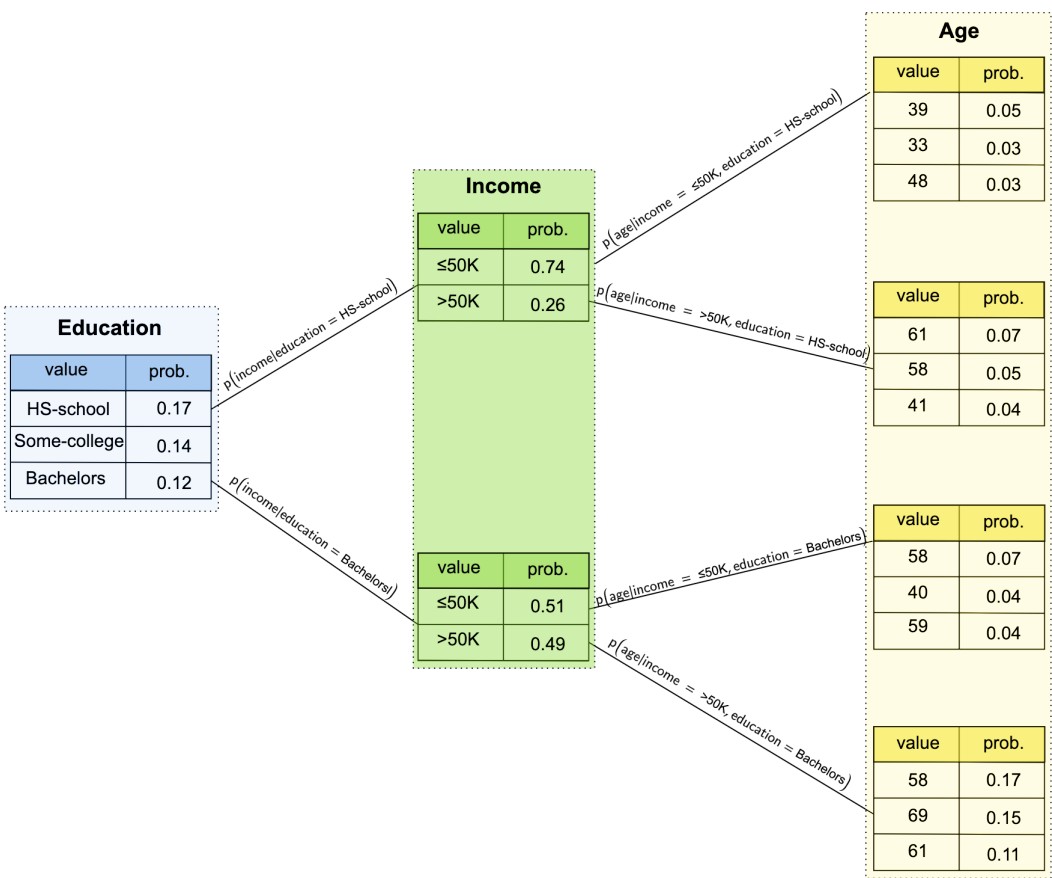

Figure 8: Example of the arbitrary conditioning using the GReaT approach on Adult data set. For this experiment, we select only three variables Education, Income, and Age from the Adult Income data set (Dua & Graff, 2017). However, the proposed method can be scaled to the arbitrary number of conditions. The results obtaining by changing the input textual sequence, e.g., Education is HS-school, Income is >50K, Age is, after we obtain the conditional discriminate distribution $p(Age|income => 50K, education = HS - school)$. Interestingly, the GReaT method successfully learned that there is only two options for the Income variable. The arbitrary sampling is supported by our Python programming framework.

