# OpenReview forum: "Language Models are Realistic Tabular Data Generators"
_ICLR.cc/2023/Conference — ICLR 2023 poster_

### Official Review · Reviewer_EaHN · 2022-10-17

**Confidence:** 3
**Correctness:** 4
**Technical Novelty And Significance:** 4
**Empirical Novelty And Significance:** 3
**Recommendation:** 8

**Clarity, Quality, Novelty And Reproducibility:**

As far as I can tell, the work is certainly novel and is an interesting and clever application of large language models. Since the approach is very straightforward, and the code is provided as an easy-to-use Python package, the reproducibility of the results seems very likely.

**Strength And Weaknesses:**

Strengths
---
GReaT operates directly on the tabular dataset, avoiding lossy data preprocessing steps such as 1) encoding of categorical data (e.g., one-hot encodings); 2) data scaling or normalization; 3) missing value imputation; and 4) outlier removal. These data preprocessing methods can often result in loss of information and ultimately reduce the quality of generated samples.

GReaT is a simple and novel approach for generating samples for tabular datasets, the most common type of data used in industry. This is a potentially significant tool that can help combat noisy or missing values, or can be used to balance imbalanced datasets.

GReaT fine-tunes a large language model using random feature order permutation, allowing sampling with arbitrary conditioning. This also allows GReaT to leverage the vast amount of contextual knowledge learned during pretraining of the large language model to better generate appropriate values for certain features given the values of other features.

In comparison with existing methods, the experimental results suggest GReaT generates samples that 1) better substitute as training data, 2) are less discernible with the actual training and test data, and 3) qualitatively produce more appropriate values than existing methods.

Weaknesses
---
Experiments are run on just 4 rather small tabular datasets, limiting claims of GReaT's generalizability.

The datasets used contain a relatively small number of features; thus, I am unsure of how GReaT would perform for much higher-dimensional tabular datasets resulting in significantly longer input sequences for fine-tuning.

One of the motivations for GReaT is the fact that tabular datasets may contain sensitive or personal information, but wouldn't a high-quality data generator potentially reproduce some of this personal data since it is trained on that information?

Minor Weaknesses
---
What is the train-test split percentage mentioned in the "Data sets." paragraph in Section 4?

In Figure 3, the occupation "Adm-clerical" in the first row of the bottom-left table does not match the first input sequence of the textual representation where the occupation is "Transport-moving".

In the paragraph "Baselines." in Section 4, "...on a only..." -> "...on only a...".

Table 1 caption, "...in case of the regression..." -> "...in the case of regression...".

In the paragraph "Baselines." in Section 4, "...an variational autoencoder..." -> "...a variational autoencoder...".

**Summary Of The Paper:**

The paper proposes GReaT (Generation of Realistic Tabular Data), a large language model fine-tuned to generate samples for tabular datasets. First, GReaT encodes examples in a tabular dataset into sequences of text, then GReaT fine tunes a large pretrained language model using the textual representation of the tabular data. Once tuned, GReaT can generate entirely new examples, or sample missing values using arbitrary conditioning.

Experiments on 4 tabular datasets (3 classification and 1 regression) demonstrate GReaT generally outperforms existing tabular data generators on a wide and comprehensive range of evaluation metrics.

**Summary Of The Review:**

GReaT leverages large language models to generate samples for tabular data in a way that avoids many of the lossy data preprocessing steps required by existing approaches, resulting in high-quality contextually-relevant tabular examples. Overall, the number of datasets evaluated is rather low, however, the positives currently slightly outweigh the negatives.

---

> ### Author Response · Authors · 2022-11-13
> **Response to Reviewer EaHN**
>
> Dear Reviewer, we thank you for appreciating the contributions of our paper and for the constructive suggestions!
>
> &nbsp;
>
> Please find below a detailed discussion of the points you have raised:
>
>
>
> >Experiments are run on just 4 rather small tabular datasets, limiting claims of GReaT's generalizability. The datasets used contain a relatively small number of features;
>
> Thank you for raising this issue. _To respond, we introduce two new real-world datasets in our work_, one with a large number of samples (more than 100,000). Both datasets come from the medical domain. Regarding the feature number the newly-added Diabetes dataset has double times more features than the largest in our initial submission 49 vs 23. *Additional datasets further confirm our findings.* In total, our work now contains 6 real-world datasets and 3 synthetic datasets, where the real-world datasets come in different sizes, reaching from less than 1,000 to over 100,000 samples. Up to 7 percent improvement on the largest data set (i.e. Diabetes) indicates that our method also works well for large samples sizes.
>
>
> > One of the motivations for GReaT is the fact that tabular datasets may contain sensitive or personal information, but wouldn't a high-quality data generator potentially reproduce some of this personal data since it is trained on that information?
>
> In our view, high-quality data generation does not necessarily imply the _exact replication_ of training samples.
> We consider high-quality data generators to be able to produce synthetic data sets which cannot be distinguished from the original test set. If there was substantial copying in the synthetic data, this information could be used to make this distinction, which requires data generators that generalize beyond training samples. We verify this definition by using two measures:
> - The distance-to-closest-record (DCR) measure makes sure none of the personal data is merely copied in the synthetic data (see Figures 4 and 5). This showed that the generated samples are close to the true samples but not exactly the same - and, therefore, not mere copies of the personal data.
> - Additionally, the discriminator metric helps to establish that the synthetic data set cannot be easily distinguished from samples from the original distribution.
>
> We would be concerned that some of the personal data could have been reproduced but we closely monitored the copying behavior of our model (see Figures 4 and 5) which strongly suggests that no copying of training data points is happening.
>
> >What is the train-test split percentage mentioned in the "Data sets." paragraph in Section 4?
>
> We split all datasets using the 80/20 percent scheme. We divided the data once to ensure that all models were trained or fine-tuned on the exact same data. We have included this information in the revised version. Thank you for your question.
>
> >In Figure 3, the occupation "Adm-clerical" in the first row of the bottom-left table does not match the first input sequence of the textual representation where the occupation is "Transport-moving".
>
> Thank you for pointing out the typos. They have been corrected. We have also proofread our paper again.
>
>
> &nbsp;
> &nbsp;
>
> We thank you for taking the time to engage thoroughly with our paper. If you have further questions or concerns, we will be happy to address them.

---

> > ### Comment · Reviewer_EaHN · 2022-11-16
> > **Response**
> >
> > I thank the authors for their response clarifying my concerns, and I have updated my score accordingly.

---

### Official Review · Reviewer_gUA4 · 2022-10-21

**Confidence:** 5
**Correctness:** 3
**Technical Novelty And Significance:** 3
**Empirical Novelty And Significance:** 2
**Recommendation:** 8

**Clarity, Quality, Novelty And Reproducibility:**

- The writing is clear, detailed and not ambiguous
- The textual encoding scheme proposed by the authors in quite novel in this setting
- The work seems easily reproducible, although the authors are strongly encouraged to share their code

Minor corrections:
- Confused Indices. In section 3.1, definition 1. the authors seem to have confused the indices for rows and columns. m was originally defined for rows, but later used for columns/features. In general, the authors are advised to clarify the notations in the various definitions
- Typos. Section 4 (baseline, line 3 and DCR, line 5)

Further clarification needed:
- The authors claim that previously proposed methods such as mode-specific normalisation introduce artificial ordering. The authors should clarify what is meant by this statement

**Strength And Weaknesses:**

Strengths:
- The proposed approach is easier to handle because it removes the bottleneck of the first preprocessing step for the numeric and categorical columns
- The textual subject-predicate-subject encoding scheme, together with the random permutation of the feature order, provides the end user with full probabilistic control over the sampling procedure of the generative model
- The paper shows significant improvements over the state of the art evaluated with the proposed methods

Weakness:
- Lack of related works. The authors claim that there are no previous works using pre-trained LLM to generate tabular data. However, a quick Google search revealed that there are some recent works (2021) that have considered a similar architecture for generating tabular data, including time series data. Granted, the pre-processing steps are different and this work is new in that regard. Regardless, the authors should highlight such related work and differentiate their work or justify the omission
- Bad samples. The authors state in Section 3.2 that some synthesised samples that violate the resulting patterns for tabular data have been discarded, but fail to provide an explanation of why such a violation occurs or provide examples of such samples to guide the reader's imagination
- Practicality: the proposed methods require significantly longer fine-tuning (~9hrs) compared to other baseline methods (< 2mins). The authors have not commented on this or justified why such a computationally intensive approach is preferable


**Summary Of The Paper:**

The paper proposes the use of large-scale autoregressive generative language models (LLMs) for generating tabular data. Several recent works have similarly attempted the task of generating tabular data using popular computer vision methods such as Variational Autoencoders and Generative Adversarial Networks, but require an initial preprocessing step of encoding the categorical columns and (optionally) transforming the numerical columns, which often have a large impact on the performance of the generative model. In this paper, a different approach is taken by using a transformer-decoder network, a popular architecture in the NLP field. In this approach, the columns are first converted to text using a textual encoding scheme, the feature order is permuted to allow arbitrary conditioning and sampling, and fine-tuning is performed using a pretrained generative LLM.

**Summary Of The Review:**

The proposed method is a breath of fresh air compared to previous work on GANs and VAEs for generating tabular data. Transformer architectures are increasingly dominating various fields, and this approach seems promising for future deep-learning-based synthesis of tabular data, but how to protect the privacy of the data synthesised from such models still needs to be studied in depth.

---

> ### Author Response · Authors · 2022-11-13
> **Response to Reviewer gUA4**
>
> Dear Reviewer, we are grateful for your thoughtful review and for recognizing our efforts!
>
> &nbsp;
> &nbsp;
>
> In the following, we provide a point-by-point response to your comments.
>
> > Lack of related works. The authors claim that there are no previous works using pre-trained LLM to generate tabular data. However, a quick Google search revealed that there are some recent works (2021) that have considered a similar architecture for generating tabular data, including time series data. Granted, the pre-processing steps are different and this work is new in that regard. Regardless, the authors should highlight such related work and differentiate their work or justify the omission
>
> Thank you for pointing that out. We have extended our search for related work. For tabular data generation with transformers, we could only find one relevant work that matches the Reviewer’s description: this is the work by Padhi et al. _on time series_ data [1]. We refer to this work in the related work section of our initial draft, but have adapted our manuscript to differentiate our work further. We note that this work is not directly comparable to our’s as it assumes sequences of dependent tabular records.
> Additionally, in the broader context of tabular data modeling, transformer models have been successfully adapted for tabular data classification [2, 3, 4] and joint modeling of tabular and textual data [5]. Still, we are unaware of any state-of-the-art tabular data generator based on this architecture or leveraging the power of recent LLMs.
> If the Reviewer has further suggestions for relevant work that should be discussed, we are more than happy to include and discuss additional references in our related work section.
>
> >Bad samples. The authors state in Section 3.2 that some synthesized samples that violate the resulting patterns for tabular data have been discarded, but fail to provide an explanation of why such a violation occurs or provide examples of such samples to guide the reader's imagination
>
> The final check for pattern matching in the generation can be seen as a means to restrict the support of our synthetic data generation to match the original data support. For example, consider the feature `Occupation` from the Adult dataset with specific support, e.g., {`Adm-clerical`, `Exec-managerial`, `Prof-specialty`, … }. In rare cases, the pre-trained large language model (LLM) might leave this support and output, for example, `Adm clerical` (without dash), or mix up these categories and return `Adm-managerial`. We drop all samples that do not exactly match the categorical support or number that cannot be parsed.
>
> The source of generating invalid samples can be identified in the sampling procedure. As random sampling (according to its conditional probability distribution) is used when sampling the next token, also tokens deemed very unlikely by the model may nevertheless be sampled from time to time. By choosing a high enough sampling temperature $T$, we can push the rate of misgeneration to be negligibly low. In practice, a simple validation function can quickly reject rare-occurring invalid samples. We have extended our manuscript with a discussion of rare-occurring invalid samples.
>
> >Practicality: the proposed methods require significantly longer fine-tuning (~9hrs) compared to other baseline methods (< 2mins). The authors have not commented on this or justified why such a computationally intensive approach is preferable
>
> In our view the increase in computational cost is well justified by the fact that our proposed method GReaT outperforms state-of-the-art models for tabular data generation: With respect to the machine learning efficiency measure (see Table 1) **we improved the best existing methods by up to 44%** (on the California Housing data set using the LR model). With respect to the discriminator measure (see Table 2), our generative model for tabular data **outperforms the state of the art by up to 30%** (on the Heloc data set).
>
> In highly critical machine learning applications (e.g., in healthcare) where the collection of new samples is usually costly or impossible due to ethical concerns, the quality of the synthetically generated samples is of great importance, and thus more computational resources to generate higher quality samples could be well justified. As an illustration, synthetic data is widely used in medical ML [1]. Therefore, if realistic synthetic data is required and computation resources are not an issue, the proposed method deserves consideration.
>
> Finally, we highlight that we will share our fine-tuned generative models on the HuggingFace model hub (https://huggingface.co/models) which make it effortless to download our fine-tuned weights with just one line of code. This avoids unnecessary fine-tuning of our method for future benchmarking.
>
>
>
> *__(continued)__*

---

> > ### Author Response · Authors · 2022-11-13
> > **Response to Reviewer gUA4**
> >
> > *__(continuation, please see the response above)__*
> >
> > ...
> >
> > >Confused Indices. In section 3.1, definition 1. the authors seem to have confused the indices for rows and columns. m was originally defined for rows, but later used for columns/features. In general, the authors are advised to clarify the notations in the various definitions
> >
> > Thank you for pointing this out; we adjusted the notation accordingly.
> >
> > >Typos. Section 4 (baseline, line 3 and DCR, line 5)
> >
> > Thank you for these catches. We have carefully (i) fixed the typos in Section 4 (ii) rechecked the notation for inconsistencies. These changes are effective in the revised version of our draft.
> >
> >
> > >The work seems easily reproducible, although the authors are strongly encouraged to share their code
> >
> > We appreciate you raising this question! We are aiming to develop an easy-to-use Python framework (with PyPi) to allow the machine learning community to synthesise state-of-the-art tabular datasets. As we mentioned before, we also have the plan to utilize the HuggingFace model hub to share our fine-tuned models, so it will be optional to redo the fine-tuning for common datasets. To conclude, we will open-source our experimental results, making them available to the community as strong benchmarks.
> >
> > &nbsp;
> > &nbsp;
> > &nbsp;
> >
> > _Dear Reviewer, We sincerely thank you for the time you spend reviewing our work_, if you have further questions or concerns, we will be happy to address them.
> >
> > &nbsp;
> > &nbsp;
> > &nbsp;
> >
> > References:
> >
> > [1]: Inkit Padhi, Yair Schiff, Igor Melnyk, Mattia Rigotti, Youssef Mroueh, Pierre Dognin Jerret Ross, Ravi Nair, Erik Altman, International Conference on Acoustics, Speech, and Signal Processing (ICASSP), 2021
> >
> > [2]: Sercan O. Arik and Tomas Pfister. TabNet: Attentive interpretable tabular learning. arxiv:1908.07442, 2019
> >
> > [3]:Gowthami Somepalli, Micah Goldblum, Avi Schwarzschild, C Bayan Bruss, and Tom Goldstein. SAINT: Improved neural networks for tabular data via row attention and contrastive pre-training. arXiv preprint arXiv:2106.01342, 2021.
> >
> > [4]: J. Kossen, N. Band, C. Lyle, A. Gomez, T. Rainforth, and Y. Gal, “Self-attention between datapoints: Going beyond individual input-output pairs in deep learning”, NeurIPS, 2021
> >
> > [5]: Pengcheng Yin, Graham Neubig, Wen-tau Yih, Sebastian Riedel, TaBERT: Pretraining for Joint Understanding of Textual and Tabular Data, Proceedings of the 58th Annual Meeting of the Association for Computational Linguistics, pages 8413–8426, 2020

---

> > > ### Comment · Reviewer_gUA4 · 2022-11-16
> > > **Further clarification needed**
> > >
> > > Dear authors,
> > >
> > > Thank you for responding to my comments and revising your manuscript. I requested further clarification which perhaps have been missed by the authors so I have provided more details below:
> > >
> > > "...For non-Gaussian feature distributions, the authors propose a mode-specific normalization technique. However, as we pointed out in Section 1, such encoding techniques can be lossy and introduce artificial orderings." - page 3
> > >
> > > It is true that categorical encoding into numeric values (using label-encoding scheme) leads to artificial ordering, as discussed in the cited work [1]. However, this is not the case with one-hot encoding which is the canonical scheme used to encode the categorical columns in most of the tabular data generative model literatures, admittedly this can lead to high-dimensional sparse feature vectors and exercerbate the "curse of dimensionality" problem in the presence of diverse set of categories in the data [1]. In addition, the paper claims that previously proposed methods such as mode-specific normalisation introduce artificial ordering. Considering that mode-specific normalisation, as introduced in [2], represents each value as a one-hot vector indicating the mode of that value, and a scalar indicating the value within the mode. It is not clear to me how this would lead to an artificial ordering. Further clarification would be appreciated.
> > >
> > > References:
> > >
> > > [1] Borisov, V., et al. "Deep neural networks and tabular data: A survey. arXiv 2021." arXiv preprint arXiv:2110.01889.
> > >
> > > [2] Xu, Lei, et al. "Modeling tabular data using conditional gan." Advances in Neural Information Processing Systems 32 (2019).

---

> > > > ### Author Response · Authors · 2022-11-16
> > > > **Response to Reviewer gUA4 (2)**
> > > >
> > > > Dear Reviewer,
> > > >
> > > > Thank you for pointing this out. We had, in fact, overlooked this point in our initial response. We agree that mode-specific normalization does not introduce artificial ordering. In the mentioned paragraph, we wanted to make the broader point that, in general, current feature preprocessing schemes lead to problems such as lossy representations and artificially introduced orderings, but also the “curse of dimensionality” problem mentioned by the Reviewer. We see that our manuscript was not precise enough at this point.
> > > >
> > > > To respond, we adjusted the manuscript by removing the sentence and introducing a new one:
> > > >
> > > > _However, the one-hot-encoding scheme used  to encode the modes and the categorical features in this work can aggravate the "curse of dimensionality" problem (Bellman,1966) in the presence of high-cardinality variables._
> > > >
> > > > We sincerely thank you again, your questions and remarks helped us to further improve our work. Please let us know if you have further questions or need additional clarification.

---

> > > > > ### Comment · Reviewer_gUA4 · 2022-11-17
> > > > > **Response**
> > > > >
> > > > > I thank the authors for the clarification and for revising their manuscript. I have no further questions.

---

### Official Review · Reviewer_HxcB · 2022-10-22

**Confidence:** 3
**Correctness:** 3
**Technical Novelty And Significance:** 3
**Empirical Novelty And Significance:** 2
**Recommendation:** 6

**Clarity, Quality, Novelty And Reproducibility:**

Hi, I'm currently on a medical leave and won't be able to perform ICRL review duties. sorry for the late notice.

**Strength And Weaknesses:**

Hi, I'm currently on a medical leave and won't be able to perform ICRL review duties. sorry for the late notice.

**Summary Of The Paper:**

Hi, I'm currently on a medical leave and won't be able to perform ICRL review duties. sorry for the late notice.

**Summary Of The Review:**

Hi, I'm currently on a medical leave and won't be able to perform ICRL review duties. sorry for the late notice.

---

> ### Author Response · Authors · 2022-11-13
> **Response to Reviewer HxcB**
>
> Dear Reviewer, we are sorry to read that you’re currently on medical leave. We hope that you get better soon and we wish you a good recovery!

---

### Official Review · Reviewer_7WMZ · 2022-11-01

**Confidence:** 2
**Correctness:** 3
**Technical Novelty And Significance:** 1
**Empirical Novelty And Significance:** 2
**Recommendation:** 5

**Clarity, Quality, Novelty And Reproducibility:**

This paper is clearly written. The text, related work, figures and tables are well prepared.

**Strength And Weaknesses:**

Strengths:

The authors proposed an interesting way to use large language models. It's a novel use case.

Weaknesses:

1. The computation cost for the proposed approach seems profitably expensive. In table 6, both GReaT and Distill-GReaT required two order of magnitudes bigger compute budget comparing to other baselines. This leads to serious scaling concerns.

2. Have the authors consider using few-shot evaluation from the large language models as the alternative way to generate synthetic tabular data? It would be cheaper than fine tuning such a big model using hundreds of epoches.  With proper prompting, large language models should be able to generate desired outputs.

3. I am not sure the significance of the proposed method. It's not clearly to me the significance of the deltas in those measures reported in the experiment section.



**Summary Of The Paper:**

This paper proposed a novel approach to generate synthetic tabular data using large language models.

**Summary Of The Review:**

The authors did find a novel application of large language models. However, it's unclear if the gains of the proposed method can justify the 100x increase in the computation cost.

---

> ### Author Response · Authors · 2022-11-13
> **Response to Reviewer 7WMZ**
>
> Dear Reviewer, we thank you for your valuable comments and the time you spent reviewing our work!
>
> &nbsp;
> &nbsp;
>
> Please find below a detailed discussion of the points you have raised:
>
> >The computation cost for the proposed approach seems profitably expensive. In table 6, both GReaT and Distill-GReaT required two order of magnitudes bigger compute budget comparing to other baselines. This leads to serious scaling concerns.
>
> In our view, the increase in computational cost is reasonably justified by multiple facts. First, our proposed method GReaT outperforms state-of-the-art models for tabular data generation: With respect to the machine learning efficiency measure (see Table 1) we **improved the best existing methods by up to 44%** (on the California Housing data set using the LR model). With respect to the discriminator measure (see Table 2), our generative model for tabular data **outperforms the state of the art by up to 30%** (on the Heloc data set).
>
> Next, in highly critical machine learning applications (e.g., in healthcare) where the collection of new samples is usually costly or impossible due to ethical concerns, the quality of the synthetically generated samples is of great importance, and thus more computational resources to generate higher quality samples could be well justified. As an illustration, synthetic data is widely used in medical ML [1]. Therefore, if realistic synthetic data is required and computation resources are not an issue, the proposed method deserves consideration.
>
> Finally, we highlight that we will share our fine-tuned generative models on the HuggingFace model hub (https://huggingface.co/models) which make it effortless to download our fine-tuned weights with just one line of code. This avoids unnecessary fine-tuning of our method for future benchmarking.
>
> Thank you for raising this question, *we added the justification of the high computation to our paper.*
>
> > Have the authors consider using few-shot evaluation from the large language models as the alternative way to generate synthetic tabular data? It would be cheaper than fine tuning such a big model using hundreds of epoches...
>
> We have tried to generate multiple samples without the pre-training steps but we quickly discovered some problems with this approach: First, the large language model could not learn the joint distributions of the tabular data with only a few examples, resulting in less realistic samples. Second, the diversity of the generated data can suffer under this few-shot-approach. Only a few samples cannot represent the whole data manifold from which we want to sample. Especially for data sets with a high number of variables, the number of shots is limited by the input size of the transformer. Therefore, the generated samples might not reconstruct the diversity of the original data – except the prompt is changed for each newly generated sample. However, this entails the need to always access the training data set during the generation process.
>
> > I am not sure the significance of the proposed method.
>
> Conceptually, we believe our work induces a fundamental paradigm shift in tabular data generation. While previously, GANs / VAEs were chosen as backbones for the task, our work is the first to suggest the use of pre-trained LLMs based on attention mechanisms (Transformers) for this task and we show that it results in state-of-the-art performance.
>
>
>
> *__(continued)__*

---

> > ### Author Response · Authors · 2022-11-13
> > **Response to Reviewer 7WMZ**
> >
> > _**(continuation, please see the response above)**_
> >
> > > It's not clearly to me the significance of the deltas in those measures reported in the experiment section.
> >
> > Every improvement to state-of-the-art tabular data generation may have a direct and significant impact in practice. For example, a company working with highly private personal data (e.g., a bank) might work with subcontractors developing data-driven software to perform fraud detection on credit card transactions. However, they cannot share their data due to privacy concerns and would like to keep it to themselves. To still allow the subcontractor to create a model for fraud detection, they can instead provide them with artificially generated data without providing their private data and which they can use to develop a suitable model.
> >
> > This exact use case is reflected by the Machine Learning Efficiency (MLE) and the Distance to closest Record (DCR) metric: While MLE reflects the accuracy of a model trained on the synthetic data when using the actual customer data, DCR measures the distances between generated samples and those used to train the generative models, ensuring the samples are not exact copies of the training data. Thus, an increase in MLE can directly be operationalized by the bank (allowing them to obtain a better model from the subcontractor), while the DCR asserts that privacy constraints are still met. _Our approach shows a superior result on the MLE measure (Tab. 1, Tab. 4), and in DCR experiments, the results display that our model generates unseen samples in the expected proximity to the original ones (Fig. 4, Fig. 5)._
> >
> > To verify that the data distribution is not misrepresented in any way, the Discriminator measure completes our set of measures by checking if the synthetic and original tabular data can be distinguished using an ML model. In this experiment, our proposed model demonstrates excellent results; it is, on average, **22% better than the best baseline model - CTGAN**. In summary, our method outperforms competitors in the sense that our data is harder to distinguish from the original data (Discriminator metric), does not contain unwanted copies (DCR), and can be used to obtain more performant models for the original data (MLE).
> >
> > &nbsp;
> > &nbsp;
> > &nbsp;
> > &nbsp;
> >
> > If you think we adequately addressed your concerns, we kindly ask you to reconsider your final score. Please let us know if you have any further questions or remarks.
> >
> > &nbsp;
> > &nbsp;
> >
> > Reference:
> >
> > [1] Hernandez, Mikel, et al. "Synthetic Data Generation for Tabular Health Records: A Systematic Review." Neurocomputing (2022). https://www.sciencedirect.com/science/article/abs/pii/S0925231222004349

---

### Author Response · Authors · 2022-11-13
**General Response to Our Submission**

Dear Reviewers, Area Chairs, and Program Chairs,

We thank the Reviewers for their valuable feedback and suggestions. Our work has been revised in response to your reviews. We highlighted changes in the manuscript using $\color{blue}{blue}$
 color, while more significant revised parts are highlighted using $\color{magenta}{magenta}$
 color.



We also want to thank the Reviewers for noting the strengths of our paper, namely:

- All participating Reviewers indicated the novel use case of pretrained large language models (LLMs) for tabular data generation. While previous works for tabular data generation mostly rely on GANs or VAEs, we demonstrate that generative LLMs are capable of producing state-of-the-art results and showing significantly better performance than GAN/VAE-based models. The proposed GReaT approach achieves superior performance on several tabular datasets across various measurements.
- Reviewers `gUA4`, `EaHN` pointed out that the proposed GReaT method removes the bottleneck of the first preprocessing step for numerical and categorical columns (e.g., one-hot encodings, data scaling or normalization, missing value imputation, outlier removal), which usually results in an information loss, and reduces the quality of generated data. `EaHN`  writes that GReaT leverages a vast amount of contextual knowledge learned during pretraining of the large language model to generate better appropriate values for certain features given the values of other features.
- Reviewers `gUA4`, `EaHN` also indicated that our approach provides full probabilistic control over the sampling procedure, whereas most other tabular data generation methods do not offer arbitrary conditioning power.
- Last but not least, Reviewer `EaHN` noted that the proposed model is a potentially significant tool that can help combat noisy or missing values or can be used to balance imbalanced datasets.

Additionally, we would like to highlight the key updates of the revised version:
- We introduce two more real-world datasets to our study. Also, with these, the proposed model demonstrates state-of-the-art performance.
- To address the common concerns of the Reviewers, we included the justification for the computation cost in our paper. In a nutshell, (1) our approach demonstrates superior performance in the vast majority of tests; for example, in the discriminator experiment (please see Tab. 2), our GReaT model shows _an average improvement of 20% across all datasets_ in our study. (2) There are critical applications, such as healthcare, where high-quality synthetic tabular data is required, regardless of the possible computation cost. (3) We are going to share our fine-tuned models online (e.g., on the HuggingFace model hub). Therefore, practitioners and researchers will not need to fine-tune the models again.
- We re-proofread our manuscript again and corrected all typos that the Reviewers pointed out. Also, we improved the consistency of our paper.
- As a final note, tabular data generation is a significant problem, and obtaining fine-grained synthetic data will be beneficial to the community.

----
In closing, we thank the Reviewers again for their time and valuable feedback. If there are further concerns, please let us know.

---

### Decision · Program_Chairs · 2023-01-20

**Decision:**

Accept: poster

**Justification For Why Not Higher Score:**

While the results are strong, the techniques are not necessarily that novel, so an ICLR audience would likely be better served by another paper.

**Justification For Why Not Lower Score:**

The method requires few tricks, the empirical results are strong, and the focus is on an underserved area.

**Metareview: Summary, Strengths And Weaknesses:**

The authors propose a way to adapt pretrained large language models to generate tabular data. The approach comprises a pre-processing of inputs to convert parts of tabular data into strings, a post-processing of the output string into tabular data, a fine-tuning the LLM, and varying the sampling strategy. While none of these methods are technically novel, the results -- especially compared to GANs and VAEs -- are strong, and merit acceptance.

**Note From Pc:**

if the above contains the word "oral" or "spotlight" please see: "oral" presentation means -> notable-top-5% and "spotlight" means -> notable-top-25%. As stated in our emails, we are disassociating presentation type from AC recommendations